# Unified picture of vibrational relaxation of OH stretch at the air/water interface

Woongmo Sung [1], Ken-ichi Inoue [1,3], Satoshi Nihonyanagi [1,2] & Tahei Tahara [1,2] ✉

The elucidation of the energy dissipation process is crucial for understanding various phenomena occurring in nature. Yet, the vibrational relaxation and its timescale at the water interface, where the hydrogen-bonding network is truncated, are not well understood and are still under debate. In the present study, we focus on the OH stretch of interfacial water at the air/water interface and investigate its vibrational relaxation by femtosecond time-resolved, heterodyne-detected vibrational sum-frequency generation (TR-HD-VSFG) spectroscopy. The temporal change of the vibrationally excited hydrogen-bonded (HB) OH stretch band ($v=1{\to}2$ transition) is measured, enabling us to determine reliable vibrational relaxation ($T_1$) time. The $T_1$ times obtained with direct excitations of HB OH stretch are 0.2-0.4 ps, which are similar to the $T_1$ time in bulk water and do not noticeably change with the excitation frequency. It suggests that vibrational relaxation of the interfacial HB OH proceeds predominantly with the intramolecular relaxation mechanism as in the case of bulk water. The delayed rise and following decay of the excited-state HB OH band are observed with excitation of free OH stretch, indicating conversion from excited free OH to excited HB OH (~0.9 ps) followed by relaxation to low-frequency vibrations (~0.3 ps). This study provides a complete set of the $T_1$ time of the interfacial OH stretch and presents a unified picture of its vibrational relaxation at the air/water interface.

Water is an essential constituent in various biological, chemical, and physical processes. Although water is the most ubiquitous liquid, it has unique properties that originate from its hydrogen bonding network. The energy of the hydrogen bond is in the order of thermal energy at room temperature[1], and hence the hydrogen bonding network of liquid water fluctuates and exchanges very rapidly, making the time scale of its vibrational dynamics sub-picosecond[2–8]. Because such ultrafast dynamics of water governs the chemical/physical properties of liquid water, it has been intensively investigated by various time-resolved vibrational spectroscopies such as IR-pump/IR-probe[4], IR-pump/Raman-probe[9,10], and 2D IR[5–7]. For example, energy dissipation, fluctuation of the hydrogen bonding network, and rotational motion of the OH moiety have been discussed based on vibrational relaxation, spectral diffusion, and anisotropy decay observed by various ultrafast spectroscopy[2,3]. Elucidation of these dynamic properties of water is critically important because it is the basis for understanding chemical and biological processes of fundamental importance in aqueous solutions.

At the surface of liquid water (the air/water interface), the hydrogen bonding network is truncated and rearranged. Therefore, a fundamental relevant question at the air/water interface is how the structure and dynamics of interfacial water differ from those of bulk water. In fact, elucidation of molecular-level properties of interfacial water is critically important to understand the mechanism of many chemical/physical processes, such as the formation of marine aerosol[11],

[1]Molecular Spectroscopy Laboratory, RIKEN, 2-1 Hirosawa, Wako, Saitama 351-0198, Japan. [2]Ultrafast Spectroscopy Research Team, RIKEN Center for Advanced Photonics (RAP), 2-1 Hirosawa, Wako, Saitama 351-0198, Japan. [3]Present address: Department of Chemistry, Graduate School of Science, Tohoku University, Sendai 980-8578, Japan. ✉e-mail: tahei@riken.jp

on-water catalysis[12–14], protein folding[15,16], ice formation[17,18], and functions of supramolecules in the aqueous phase, for which interfacial water plays vital roles[19]. Despite its importance, the ultrafast dynamics of water at interfaces is still largely veiled because of the difficulty in selectively probing interfacial water molecules in a few nanometers of thickness.

Thanks to the intrinsic interface selectivity, even-order nonlinear optical techniques have been successfully applied to experimental studies of the structure and dynamics at the air/water interface as well as other aqueous interfaces. In particular, vibrational sum-frequency generation (VSFG)[20–22] spectroscopy, including heterodyne-detected (HD-) VSFG[22–27], has been intensively utilized for clarifying the structure of interfacial water. Conventional VSFG can only provide spectra of a modulus square of second-order susceptibility $|\chi^{(2)}|^2$. Therefore, it cannot directly provide the frequency of a vibrational resonance due to interference with the nonresonant background and cannot give the sign information of $\chi^{(2)}$ that is essential for determining the absolute orientation of interfacial water. In contrast, HD-VSFG enables direct measurements of the $\chi^{(2)}$ spectra with their sign, allowing us to determine the absolute orientation of the interfacial water molecules. More importantly, HD-VSFG provides the spectrum of the imaginary part of $\chi^{(2)}$ (Im$\chi^{(2)}$) that is free from spectral distortion, and hence they can be interpreted straightforwardly as in the case of IR absorption/Raman spectra of bulk[28].

The vibrational dynamics of water at aqueous interfaces was initially investigated by time-resolved VSFG (TR-VSFG) with conventional homodyne detection[29–32]. However, homodyne TR-VSFG does not provide the direct transient response of $\chi^{(2)}$ because the observed pump-induced change of the SFG intensity corresponds to a complex quantity of the steady-state $\chi^{(2)}$ and transient signal $\Delta\chi^{(2)}$ (2Re($\chi^{(2)}$ $\Delta\chi^{(2)}$) + $|\Delta\chi^{(2)}|^2$)[28,33,34]. Therefore, analysis of homodyne TR-VSFG spectra is complicated, very often making it difficult to derive correct information on interfacial dynamics from the data obtained. On the other hand, time-resolved heterodyne-detected VSFG (TR-HD-VSFG) provides $\Delta\chi^{(2)}$[28,33–35], and $\Delta$Im$\chi^{(2)}$ spectra can be directly compared to the time-resolved infrared or Raman spectra that correspond to $\Delta$Im$\chi^{(1)}$ and $\Delta$Im$\chi^{(3)}$ spectra, respectively. Furthermore, TR-HD-VSFG measurements carried out with various pump frequencies can provide 2D spectra, i.e., two-dimensional (2D) HD-VSFG spectra[34,36], which are interface counterparts of 2D-IR spectra[33,34,36–40].

The first TR-HD-VSFG (2D HD-VSFG) study at the air/water interface was reported by our group, which was carried out for the OH stretch region covering both the hydrogen-bonded (HB) OH and free OH[34]. Since then, several TR-HD-VSFG and 2D HD-VSFG studies have been performed for unveiling the ultrafast vibrational dynamics of the OH stretch vibration at the air/water interface[41–43]. Bonn and coworkers performed 2D HD-VSFG[41] and TR-HD-VSFG[42] experiments and reported a marked pump frequency dependence of the spectral diffusion and vibrational relaxation ($T_1$) time of HB OH stretch: They reported 2D HD-VSFG spectra exhibiting a very slow spectral diffusion (-1.5 ps) on the high-frequency side of the HB OH band and argued that the HB OH stretch around 3500 cm$^{-1}$ is energetically isolated at the air/water interface[41]. In the following TR-HD-VSFG study, they also reported a significant pump frequency dependence of the $T_1$ time of the OH stretch, including a surprisingly long $T_1$ time (-750 fs) at 3500 cm$^{-1}$[42]. However, our TR-HD-VSFG measurements carried out with two different pump frequencies showed that the spectral diffusion on both the low- and high-frequency sides of the HB OH band proceeds very rapidly on a few hundred femtosecond time scale[43]. It was also shown that the slow spectral diffusion reported in the previous study was an artifact that arose from a difference in the final local temperature due to the high pump pulse energy adopted in the measurements[43]. These initial confusions emerged because the bleach signal of the HB OH band, which was overlapped with the thermalized signal, was measured and analyzed without sufficient care.

Vibrational relaxation dynamics of the free OH has been successfully clarified by a recent TR-HD-VSFG study that tracked bleach recovery of the free OH band. It was revealed that the $T_1$ time of the free OH is insensitive to isotopic dilution, indicating that the major vibrational relaxation pathway of the free OH is its reorientational motion that proceeds on a -0.9 ps time scale[44]. This conclusion is in sharp contrast to that of a prior homodyne TR-VSFG study which proposed that intramolecular energy transfer was the major relaxation pathway[45]. The reorientational relaxation mechanism of the excited free OH was further confirmed by the most recent TR-HD-VSFG study on the vibrational relaxation of free OD, which showed that the vibrational relaxation of the free OD proceeds on a much slower -1.7 ps time scale, being insensitive to the isotopic dilution[46]. These studies imply that the reorientational relaxation of the excited free OH generates the excited HB OH. However, the transient signal corresponding to the converted excited HB OH stretch has not been directly observed because the monitored bleach region is overlapped by the signals due to the excited-state band ($v = 1 \rightarrow 2$ transition) of the free OH stretch as well as the frequency-shifted ground-state HB OH band that appears due to the anharmonic coupling[44].

As described above, the information of the most fundamental dynamic quantity at the air/water interface, i.e., vibrational relaxation $T_1$ time of the OH stretch of interfacial water, is gathering piece by piece. However, an overall picture has not been acquired yet. To obtain a clear, consistent view of the vibrational relaxation of water at the air/water interface, it is critically important to selectively probe a frequency region that directly provides unambiguous information on the lifetime of the vibrationally excited state of the OH stretch, being free from the disturbance of the thermalized signal and spectral diffusion.

In the present study, we carry out comprehensive TR-HD-VSFG measurements on the vibrational dynamics of the OH stretch at the air/water ($H_2O$) interface by probing the excited-state HB OH band. With pump frequencies covering HB OH and free OH stretch in the entire OH stretch region, the pump-frequency-dependent dynamics of the excited HB OH is examined. The obtained results reveal the overall relaxation dynamics of the OH stretch at the air/water interface and clarify their difference from and similarity to the relaxation dynamics in liquid water in the bulk.

## Results

### Steady-state Im$\chi^{(2)}$ spectrum at the air/water interface

Figure 1a shows the Im$\chi^{(2)}$ spectrum of the air/water interface obtained by HD-VSFG spectroscopy, which exhibits negative and positive OH stretch bands[24]. The broad negative band expanding from 3100 to 3600 cm$^{-1}$ is assigned to the stretch vibration of the HB OH moieties at the interface, whereas the sharp band peaked at 3700 cm$^{-1}$ is due to the stretch vibration of the free OH of the topmost water layer[47]. The negative and positive signs of these bands indicate "H-down" (hydrogen is pointing into bulk) and "H-up" (pointing into the air) orientation of the corresponding OH moieties, respectively. The center frequency and bandwidth of the HB OH band of the air/water interface are similar to those in the IR absorption spectrum of bulk water[48], indicating that the averaged hydrogen bonding strength of the interfacial water is comparable to that of bulk water. On the other hand, the free OH band is observed only at the interface and is characteristic. The amplitude of the free OH band is not very different from that of the HB OH band, reflecting that only a few water layers at the interface mainly contribute to the HB OH band. In fact, molecular dynamics simulation indicated that the HB OH band is dominated by the contribution from the first 2–3 water layers at the air/water interface[49,50].

Figure 1b shows the spectra of IR pump pulses used in the TR-HD-VSFG experiments. The IR pump pulses centered at 3200 (red), 3300 (orange), 3400 (green), and 3500 cm$^{-1}$ (cyan) excite the HB OH stretch. The IR pump pulse at 3700 cm$^{-1}$ (purple) excites the free OH stretch exclusively. The IR pump at 3600 cm$^{-1}$ (blue) is close to the zero-

crossing point between the negative HB OH and positive free OH bands, and hence it simultaneously excites the free OH and the high-frequency side of the broadly distributed HB OH stretch. Using these IR pump pulses with six different frequencies, all the OH stretch vibrations at the air/water interface can selectively be excited.

In the present TR-HD-VSFG study, we focus on the population dynamics of the excited HB OH stretch and determine the reliable vibrational relaxation $T_1$ time of the interfacial OH stretch. As mentioned in the introduction, the bleach of the HB OH band is overlapped

by the thermalized signal that is not directly related to the population of excited-state HB OH. An alternative way to determine the $T_1$ time is to analyze the temporal change of the excited-state band ($v = 1 \rightarrow 2$ transition) of the HB OH stretch. The large anharmonicity of the HB OH stretch causes a substantial redshift of the excited-state band, and the HB OH stretch after thermalization appears on the opposite higher frequency side. Therefore, it is expected that the thermalized signal does not noticeably affect the transient signal in the excited-state HB OH band region. Thus, the excited-state band of the HB OH stretch allows us to selectively observe the population change of the excited HB OH at the air/water interface and hence to determine the reliable vibrational relaxation $T_1$ time.

### Time-resolved $\Delta Im\chi^{(2)}$ spectra in the excited-state band region of the HB OH stretch

Figure 2a-d shows the time-resolved $\Delta Im\chi^{(2)}$ spectra in the excited-state HB OH band region at the air/water interface, which were measured with exciting different parts of the broad steady-state HB OH stretch band ($\omega_{pump}$: 3200–3500 cm$^{-1}$). In these $\Delta Im\chi^{(2)}$ spectra, a broad negative band centered around 3100 cm$^{-1}$ appears immediately after excitation ($T = 0.0$ ps), regardless of the difference in the pump frequency. This instantaneous rise indicates that the negative band originates from the excited state of the HB OH stretch that is directly generated by the IR pump pulse, assuring that the excited-state band of HB OH is successfully observed. The center frequency of the excited-state band differs from that of the steady-state HB OH band (Fig. 1a) by ~350 cm$^{-1}$ due to the large anharmonicity of the OH stretch vibration. This shift is similar to that of bulk water (~300 cm$^{-1}$)[51]. The amplitude of the excited-state band signal reaches the maximum at $T = 0.1$ ps, reflecting the finite time resolution (~200 fs) of the present measurements, and then it decays on a sub-picosecond time scale. The $\Delta Im\chi^{(2)}$ spectra at $T = 1.0$ ps and 1.5 ps show no significant difference in amplitude and spectral shape. It indicates that the air/water interface

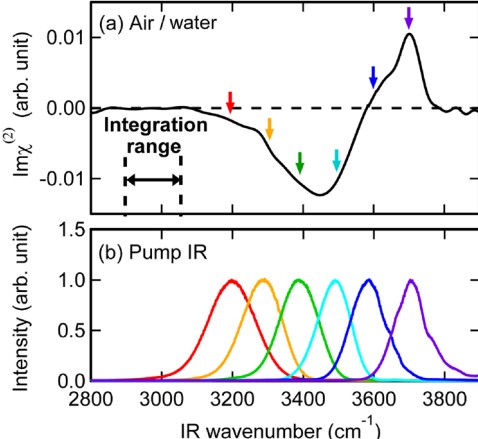

**Fig. 1 | Steady-state Imχ$^{(2)}$ spectrum of the air/water interface and spectra of IR pump pulses. a** Steady-state Imχ$^{(2)}$ spectrum of the air/water interface in the OH stretch region (solid black line). **b** Spectra of IR pump pulse used in the TR-HD-VSFG experiment. The pump IR spectra are obtained from sum-frequency signal generated at the air/quartz interface. Six colored arrows in (**a**) indicate the center frequencies of the pump IR spectra shown in (**b**).

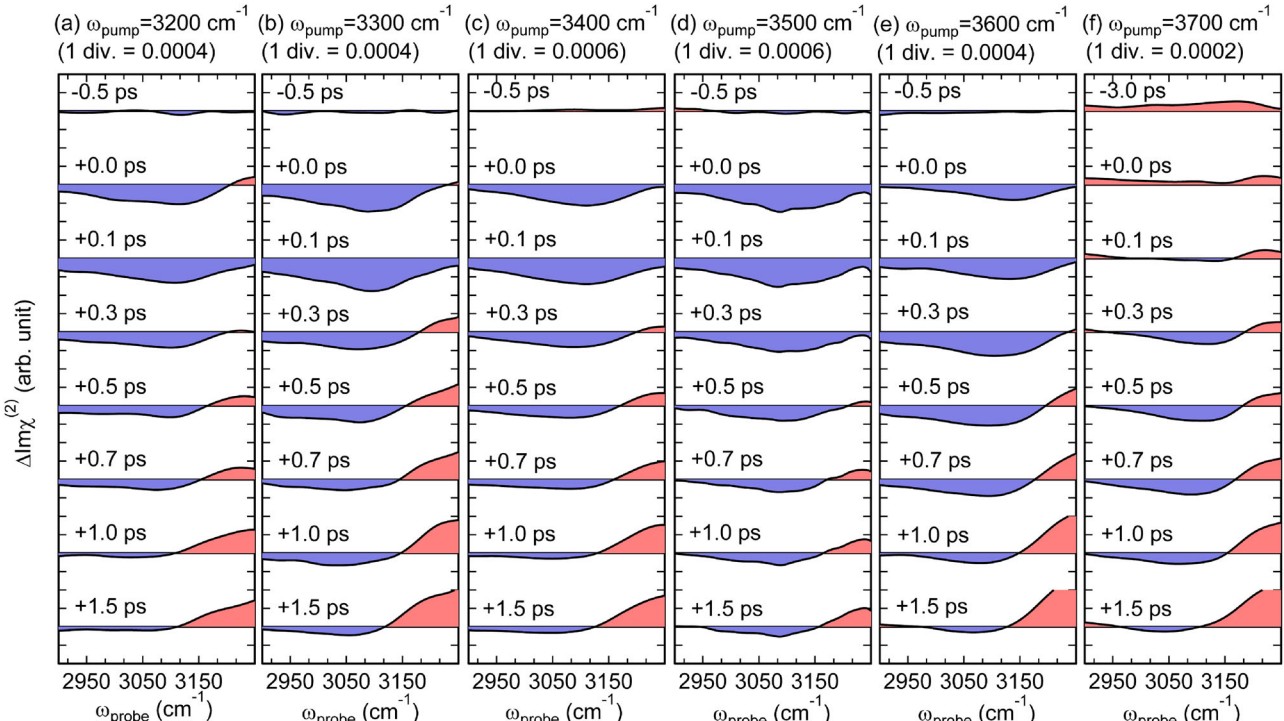

**Fig. 2 | Time-resolved $\Delta Im\chi^{(2)}$ spectra at the air/water interface in the 2900–3250 cm$^{-1}$ region.** The $\Delta Im\chi^{(2)}$ spectra are obtained with six pump frequencies at (**a**) 3200 cm$^{-1}$, (**b**) 3300 cm$^{-1}$, (**c**) 3400 cm$^{-1}$, (**d**) 3500 cm$^{-1}$, (**e**) 3600 cm$^{-1}$, and (**f**) 3700 cm$^{-1}$. The $\Delta Im\chi^{(2)}$ spectra are measured at pump-probe delays from

negative delay time (−0.5 ps or −3.0 ps) to positive delay time of +1.5 ps. The red and blue shaded areas indicate positive and negative signs of the $\Delta Im\chi^{(2)}$ spectra, respectively. 1 div. indicates 1 division of the y-axis.

reaches a quasi-thermal equilibrium condition within 1 ps, which is consistent with our previous study on the ground-state bleach (GSB) region of the HB OH stretch[43]. At a glance, no noticeable difference is seen in the decay dynamics of the excited-state HB OH band measured with the four different pump frequencies.

Figure 2f shows the time-resolved $\Delta \text{Im}\chi^{(2)}$ spectra obtained with the free OH excitation ($\omega_\text{pump} = 3700$ cm$^{-1}$). A marked difference from the spectra obtained with the HB OH excitation is that the $\Delta \text{Im}\chi^{(2)}$ spectrum shows no negative excited-state HB OH stretch band at $T = 0$ ps beyond the noise level. This absence of the excited-state HB OH stretch band immediately after excitation is natural because the 3700-cm$^{-1}$ pump pulse excites the free OH, not HB OH directly. Interestingly, however, the excited-state HB OH stretch band starts to appear at around $T = 0.3$ ps, and its amplitude reaches the maximum at around $T = 0.7$ ps. This delayed appearance of the excited-state HB OH band is direct experimental evidence for the conversion from the excited-state free OH to the excited-state HB OH. After $T = 0.7$ ps, the negative excited-state HB OH band starts to decay.

Figure 2e shows the time-resolved $\Delta \text{Im}\chi^{(2)}$ spectra obtained with the 3600-cm$^{-1}$ excitation. This pump pulse excites both the blue edge of the HB OH band and the free OH band. Compared to the data taken with excitation of HB OH ($\omega_\text{pump} \le 3500$ cm$^{-1}$; Fig. 2a–d), the $\Delta \text{Im}\chi^{(2)}$ spectra with the 3600-cm$^{-1}$ pump exhibit a slower rise of the excited-state HB OH band, which gets maximized at $T = 0.3$ ps. This temporal change of the $\Delta \text{Im}\chi^{(2)}$ spectra looks like a mixture of those observed with HB OH excitation and free OH excitation. We will discuss this observation in more detail later, based on the fitting analysis.

In all the time-resolved $\Delta \text{Im}\chi^{(2)}$ spectra shown in Fig. 2, a positive band appears in the higher frequency side above 3100 cm$^{-1}$ at delay times later than 0.3–0.5 ps. (Note that this spectral change is seen as an additional increase of the positive $\Delta \text{Im}\chi^{(2)}$ signal in the case of free OH excitation, as shown in Fig. 2f.) These positive bands are attributable to the thermalized signal, which appears with the rise of the local temperature: As the OH stretch vibrational energy of water molecules is dissipated to the surroundings, the local temperature rises and the net strength of hydrogen bonding of interfacial water is weakened[4,5,34,43]. As a result, the HB OH band is blue-shifted, generating a positive band in its low-frequency side (<3500 cm$^{-1}$) and a negative band in its high-frequency side (>3500 cm$^{-1}$) in the $\Delta \text{Im}\chi^{(2)}$ spectra[43]. We note that such a thermalized signal is expected not to appear in the frequency region lower than the HB OH stretch band in the steady-state $\text{Im}\chi^{(2)}$ spectrum. In other words, the transient $\Delta \text{Im}\chi^{(2)}$ spectra below 3100 cm$^{-1}$ are free from the thermalized signal and solely attributable to the excited-state HB OH band. Therefore, we can selectively obtain information on the temporal change of the excited-state population of the HB OH stretch from the analysis of time-resolved $\Delta \text{Im}\chi^{(2)}$ spectra below 3100 cm$^{-1}$, as described in the next two sections.

### Temporal profile observed with excitation of the HB OH stretch (3200–3500 cm$^{-1}$)

To quantitatively examine the relaxation dynamics of the excited state of HB OH, we integrate the excited-state signal in the frequency region of 2090–3050 cm$^{-1}$, which is free from the thermalized signal. Note that we did not perform the Fresnel factor correction for this analysis, because it does not affect the $\text{Im}\chi^{(2)}$ spectra below 3100 cm$^{-1}$ (see Supplementary Note 7 in Supplementary Information). Figure 3 shows the temporal profiles of the (integrated) amplitude of the excited-state HB OH signals observed with direct excitations of the HB OH stretch ($\omega_\text{pump}$: 3200–3500 cm$^{-1}$). In common, these traces exhibit instantaneous rise and sub-ps decay, representing the population change of the first vibrationally excited state of the HB OH. The signal amplitude becomes constant after 1 ps and persists up to ~100 ps (See Supplementary Fig. 2 in Supplementary Information). Such a long-lived component was also observed in a previous homodyne TR-VSFG study,

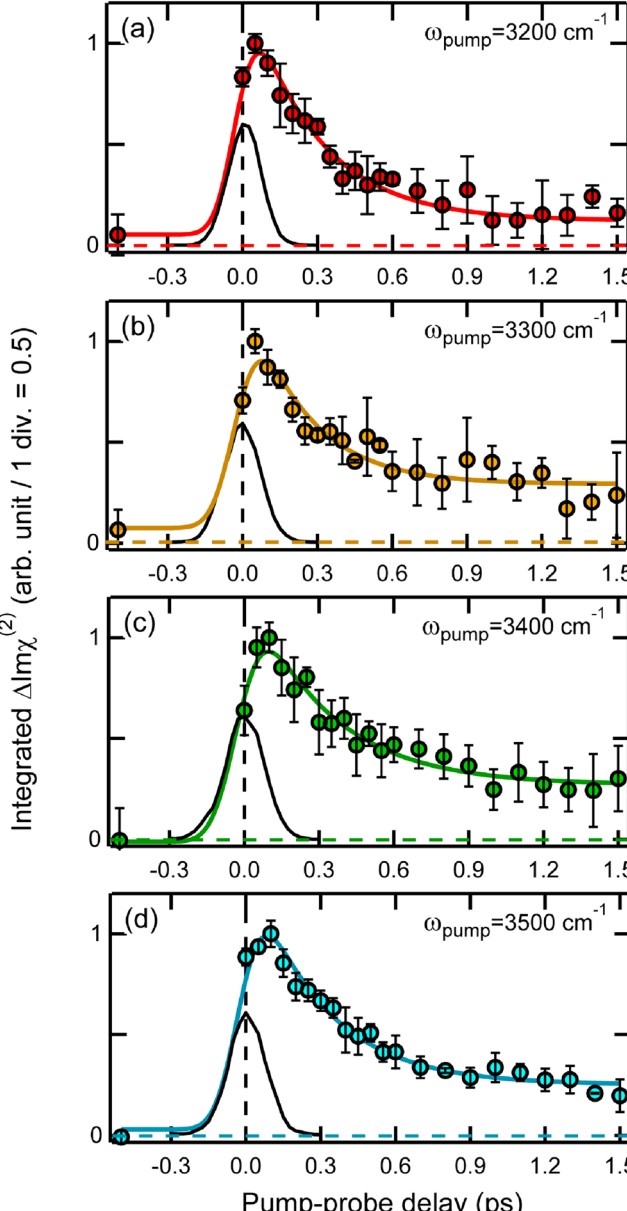

**Fig. 3 | Temporal traces of the $\Delta \text{Im}\chi^{(2)}$ signals due to the excited-state HB OH stretch at the air/water interface observed upon direct excitations of the HB OH stretch band.** The temporal traces are obtained with excitation frequencies ranging from 3200 to 3500 cm$^{-1}$. The amplitude of the profiles are normalized to unity and the sign is reversed for clear view. The pump frequency is shown in each panel. The integration range for (**a**) to (**d**) is 2900–3050 cm$^{-1}$. The colored solid lines are the best fits using Eq. S1 in Supplementary Information. The black solid lines are the instrumental responses which were measured as the cross correlation between the pump IR and the probe IR pulses (-160–190 fs full-width at half-maximum). The error bars indicate the standard deviation of the integrated $\Delta \text{Im}\chi^{(2)}$ obtained in several experiments.

and it was attributed to a water density gradient near the interface which is caused by the IR excitation[52]. Although the origin of this long-lived component is not very clear at the moment, we can consider that it does not represent any population dynamics of the excited-state HB OH because it remains even after the thermalization process completes.

With the excitation at 3200–3500 cm$^{-1}$, the excited state of HB OH stretch is directly generated, and it decays with the vibrational relaxation process. Therefore, its relaxation scheme can be

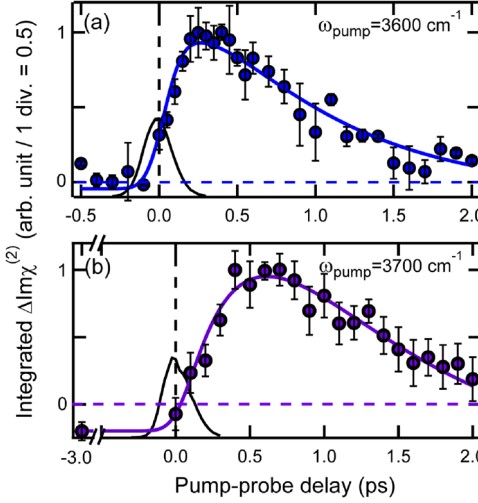

**Fig. 4 | Temporal traces of the ΔImχ$^{(2)}$ signals due to the excited-state HB OH stretch at the air/water interface observed upon excitation of free OH stretch.** The temporal traces are obtained with pump frequencies of 3600 and 3700 cm$^{-1}$. The amplitude of the profiles are normalized to unity and the sign is reversed for clear view. The pump frequency is shown in each panel. The integration range for (**a**) is 2900–3050 cm$^{-1}$ while that for (**b**) is 2950–3100 cm$^{-1}$. The colored solid lines are the best fits using Eq. S2 in Supplementary Information. The black solid lines are the instrumental responses which were measured as cross correlation between the pump IR and the probe IR pulses (~210 fs full-width at half-maximum). The error bars indicate the standard deviation of the integrated ΔImχ$^{(2)}$ obtained in several experiments.

represented as,

$$\text{HB OH} \xrightarrow{h\nu} \text{HB OH*} \xrightarrow{k_{1,HB}} \text{Intermediate*}, \tag{1}$$

where HB OH, HB OH*, and Intermediate* are the ground state of the HB OH stretch, the excited state of the HB OH stretch, and other excited vibrational states that accept the vibrational energy of the excited state of HB OH, respectively. In the case of bulk water, it has been indicated that the excited state of the HB OH stretch relaxes to low-frequency vibrations such as HOH bending and libration[53,54] before the system reaches the thermalized state that corresponds to an increase in the local temperature[55]. Using a decay rate constant $k_{1,HB}$ (1/$T_{1,HB}$), the population change of the excited HB OH can be represented as,

$$N_{HB}(t) = N_{0,HB} \exp\left(-\frac{t}{T_{1,HB}}\right), \tag{2}$$

where $N_{0,HB}$ and $T_{1,HB}$ are the initial population of the first excited state of the HB OH stretch and its lifetime ($T_1$), respectively.

The fitting of the temporal profiles shown in Fig. 3 was performed with Eq. (2) plus offset and a long-lasting component, taking account of the instrumental response of each measurement. We introduced the offset and a long-lasting component because the signal at negative time delay is not exactly zero due to the noise and the long constant signal appears at long delay times as already mentioned. (See Supplementary Note 2 in Supplementary Information for the details). The best fits obtained are shown in Fig. 3 with solid lines, and the parameters of the best fits are listed in Supplementary Table 2 in Supplementary Information. As seen, the fitted curves well reproduce the observed temporal profiles and provide the $T_{1,HB}$ values of 0.28 ± 0.03, 0.24 ± 0.05, 0.32 ± 0.06, and 0.29 ± 0.04 ps for the pump frequencies at 3200, 3300, 3400, and 3500 cm$^{-1}$, respectively.

To check the robustness of the fitting analysis, we also carried out a singular value decomposition (SVD) analysis of the ΔImχ$^{(2)}$ spectra.

The SVD analysis provided the $T_{1,HB}$ values of 0.20 ± 0.02, 0.29 ± 0.03, 0.34 ± 0.03, and 0.33 ± 0.03 ps for the 3200, 3300, 3400, and 3500-cm$^{-1}$ excitations, respectively. These values agree well with the values obtained by the fitting analysis of the excited-state HB OH signals, except for the case of the 3200-cm$^{-1}$ excitation where the SVD analysis is perturbed by spectral diffusion observed in the early time (See Supplementary Note 4 in Supplementary Information for the details).

**Temporal trace observed with excitation of the free OH stretch (3600 and 3700 cm$^{-1}$)**

Figure 4 shows temporal traces of the excited-state HB OH signals that were observed with pump pulses at 3600 and 3700 cm$^{-1}$. As the steady-state Imχ$^{(2)}$ spectrum in Fig. 1 shows, these pump pulses can excite the free OH stretch. The temporal traces of the excited-state HB OH signal observed in this case are significantly different from those observed with direct excitation of HB OH. In particular, in the temporal trace obtained with 3700-cm$^{-1}$ excitation, the excited-state HB OH signal clearly shows a delayed rise and a much slower decay. This delayed appearance of the excited-state HB OH signal directly indicates that it takes time to populate the excited state of the HB OH stretch when we initially excite the free OH at the interface. This observation is fully consistent with our previous TR-HD-VSFG study on the vibrational relaxation of the free OH, which concluded that the excited free OH is converted to the excited HB OH through the reorientational motion of the OH moiety[44]. Therefore, the relevant relaxation scheme is represented as follows,

$$\text{Free OH} \xrightarrow{h\nu} \text{Free OH*} \xrightarrow{k_{1,free}} \text{HB OH*} \xrightarrow{k_{1,HB}} \text{Intermediate*}. \tag{3}$$

Here, Free OH, Free OH* are the ground and excited states of the free OH, respectively. HB OH* is the excited state of the HB OH stretch, and Intermediate* is the excited state of other vibrational states that accept the vibrational energy of the excited state of HB OH, as in the case of the scheme (1). Then, the population changes of the excited states of the free OH ($N_{free}$) and the HB OH ($N_{HB}$) are given as follows:

$$N_{free}(t) = N_0 \exp\left(-\frac{t}{T_{1,free}}\right), \tag{4}$$

$$N_{HB}(t) = N_0 \frac{T_{1,HB}}{T_{1,HB} - T_{1,free}} \left[-\exp\left(-\frac{t}{T_{1,free}}\right) + \exp\left(-\frac{t}{T_{1,HB}}\right)\right], \tag{5}$$

Here, $T_{1,free}$ (1/$k_{1,free}$) and $T_{1,free}$ (1/$k_{1,HB}$) are the decay time constants ($T_1$'s) of the excited states of the free OH and HB OH, respectively[30].

In the actual experiments, particularly those with 3600-cm$^{-1}$ excitation, the IR pump pulse excites not only the free OH band but also the high-frequency side of the broad HB OH band. Therefore, the population of the excited-state HB OH is considered to show the kinetics that originates from both free OH excitation and direct HB OH excitation:

$$N_{HB}(t) = N_0(1-r) \frac{T_{1,HB}}{T_{1,HB} - T_{1,free}} \left[-\exp\left(-\frac{t}{T_{1,free}}\right) \right.$$
$$\left. + \exp\left(-\frac{t}{T_{1,HB}}\right)\right] + N_0 r \exp\left(-\frac{t}{T_{1,HB}}\right). \tag{6}$$

Here, $r$ represents the fraction of the HB OH stretch that is directly excited by the IR pump pulse. Note that Eq. (6) converges to Eq. (5) when $r = 0$, while it converges to Eq. (2) when $r = 1$.

The temporal traces of the excited-state HB OH signals obtained with 3600- and 3700-cm$^{-1}$ excitations were globally fitted with Eq. (6) plus an offset and a long-lasting component as in the case of the fitting for the direct HB OH excitation. The best fits are shown with solid lines

in Fig. 4a,b, and the parameters of the best fits are listed in Supplementary Table 3 in Supplementary Information. These fits well reproduce the observed temporal traces and provide $T_{1,HB} = 0.34 \pm 0.06$ ps and $T_{1,free} = 0.84 \pm 0.08$ ps. We note that the temporal profile obtained with 3700-cm$^{-1}$ excitation is fitted well with $r = 0$, implying that the 3700-cm$^{-1}$ pump pulse selectively excites the free OH.

To check the robustness of the fitting analysis, we also carried out a singular value decomposition (SVD) analysis of the time-resolved $\Delta Im\chi^{(2)}$ spectra obtained with 3600- and 3700-cm$^{-1}$ excitations. The SVD analysis provided the $T_{1,HB}$ and $T_{1,\ free}$ times of $0.37 \pm 0.06$ ps and $0.82 \pm 0.20$ ps for the 3600-cm$^{-1}$ excitation, and $0.33 \pm 0.20$ ps and $0.97 \pm 1.17$ ps for the 3700-cm$^{-1}$ excitation, respectively. These values agree well with the values obtained by the fitting analysis of the excited-state HB OH signals within the error. (See Supplementary Note 4 in Supplementary Information for the details)

It is noteworthy that the excited-state HB OH signal exhibits a counter-intuitive temporal change with 3700-cm$^{-1}$ excitation. The relaxation scheme (3) intuitively makes us anticipate that the excited-state HB OH signal appears with $T_{1,free}$ ($1/k_{1,free}$) and decays with $T_{1,HB\ OH}$ ($1/k_{1,\ OH}$). However, the observed temporal change is the opposite: the excited-state HB OH signal appears with $T_{1,HB}$ (0.34 ps) and decays with $T_{1,free}$ (0.84 ps), as shown in Fig. 4b. This temporal behavior of the excited-state HB OH signal looks peculiar, but it is a straightforward consequence of the formula (5): In the case that $T_{1,free} > T_{1,HB\ OH}$, the pre-exponential factor becomes negative, so the signal appears with $T_{1,HB}$ and decays $T_{1,\ free\ OH}$. We note that this counter-intuitive dynamics simply arises from the mathematical solution of a rate equation based on Scheme (3).

## Frequency dependence of the $T_1$ time of the OH stretch at the air/water interface

Figure 5 plots the pump frequency dependence of the $T_1$ times of the OH stretch at the air/water interface, which was determined in this study by the TR-HD-VSFG measurements of the excited-state band of the HB OH stretch. The excitations in the range of 3200–3500 cm$^{-1}$ directly excite the HB OH and provide the $T_1$ time of excited HB OH (red circle), whereas the excitations at 3600 and 3700 cm$^{-1}$ excite the free OH and provide the $T_1$ times of both the free OH (blue square) and HB OH (red circle). For comparison, the $T_1$ times of the OH stretch in bulk water reported previously are also plotted: Purple triangles depict

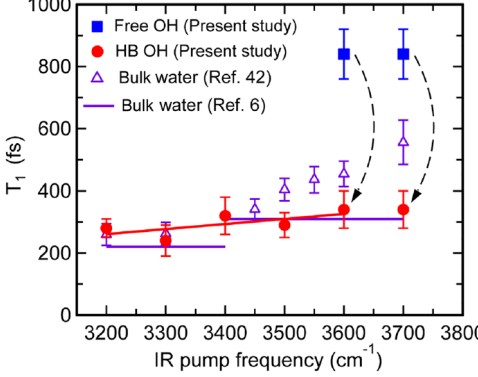

**Fig. 5 | Frquency dependence of the $T_1$ time of OH stretch at air/water interface and comparison with the values in bulk water.** $T_1$ time of the HB OH (red circle) and the free OH stretch (blue square) at the air/water interface determined by the decay of the excited-state HB OH stretch signal. The solid red line is an eye guide. $T_1$ times of bulk water from ref. 42 (purple triangle) and ref. 6 (purple bars) are also plotted for comparison. Black curved arrows connecting the blue square and red circle represent the conversion from the excited free OH to the excited HB OH stretch, which is observed with excitation at 3600 and 3700 cm$^{-1}$. The error bars for the interfacial $T_1$ time indicate the fitting errors for each value. (See Supplementary Tables 2 and 3 of Supplementary Information.).

the values determined by the IR pump-probe experiments[42], whereas the purple bars indicate the values below and above 3400 cm$^{-1}$ evaluated in a 2D IR study[6]. This plot clearly shows that the $T_1$ time of the excited HB OH at the air/water interface is not sensitive to the pump frequency, even if it might slightly become larger as the frequency becomes higher. This insensitive frequency dependence of the $T_1$ time of HB OH at the interfacial water, as well as the $T_1$ values themselves, is very similar to the HB OH in the bulk water, and in particular, the $T_1$ time is essentially the same as the bulk value determined by the 2D-IR study[6]. On the other hand, the $T_1$ time of the free OH is largely different, and it is ~3 times longer than that of HB OH of the bulk water.

The vibrational relaxation $T_1$ time of the interfacial HB OH stretch determined in this study shows a stark difference from the conclusion of a previous study of the Bonn group: they claimed that the HB OH stretch at the air/water interface exhibits much slower vibrational relaxation compared to that of bulk water[42]. In particular, they reported that the $T_1$ time of excited HB OH showed a drastic excitation frequency dependence in the high-frequency region and that the $T_1$ time of HB OH stretch at 3500 cm$^{-1}$ becomes as long as ~750 fs. These conclusions of the previous study are clearly denied by the present TR-HD-VSFG measurements on the excited-state band region of the HB OH stretch, which is free from the disturbance from the thermalized signal. In the previous study, they evaluated the $T_1$ time of HB OH stretch at 3500 cm$^{-1}$ from the temporal change of the excited-state signal around 3200 cm$^{-1}$ to minimize the influence of the thermalized signal. As clearly seen in Figs. 1a and 2d, however, the steady-state Im$\chi^{(2)}$ signal is not zero at 3200 cm$^{-1}$, and hence the time-resolved $\Delta Im\chi^{(2)}$ spectra in this frequency region are still substantially affected by the thermalized signal. Thus, the long $T_1$ time and surprisingly slow vibrational relaxation reported previously for the air/water interface are considered to arise from the effect of the thermalized signal. By the present TR-HD-VSFG measurements performed with high phase accuracy and selection of a proper probing frequency region, it is now clear that there is no peculiar slow vibrational relaxation of the HB OH stretch at the air/water interface.

The $T_1$ values of the HB OH stretch at 3500 cm$^{-1}$ ($0.29 \pm 0.04$ ps) determined in the present study agree well with the value ($0.35 \pm 0.03$ ps) evaluated in our previous TR-HD-VSFG study on the ground-state bleach (GSB) carried out with special care on the pump energy[43]. Nevertheless, the $T_1$ time at 3300 cm$^{-1}$ ($0.24 \pm 0.05$ ps) slightly deviates from the value obtained from the GSB measurement ($0.39 \pm 0.04$ ps)[43]. This deviation probably arises from the spectral diffusion of the GSB band. Indeed, the GSB band of the HB OH stretch generated with the 3300-cm$^{-1}$ excitation noticeably shifts in time due to the spectral diffusion on this time scale[43]. Therefore, we consider that the $T_1$ time determined in the present study is more reliable. It would be noteworthy that the same group of ref. 42. measured $T_1$ time of the HB OH stretch at air/water interface using homodyne TR-VSFG experiment[56] and reported that the $T_1$ time at 3310 cm$^{-1}$ excitation is ~110 fs which is about one-third of the value claimed in ref. 42 Although their new experiment was performed at 278 K, this large deviation in the $T_1$ time value demonstrates the difficulty of homodyne TR-VSFG in the determination of reliable information because it detects the product of steady-state and transient responses, making analysis of time-resolved data very complicated.

## Discussion

For the HB OH, the present study reveals that the vibrational relaxation $T_1$ time of interfacial water is very similar to that of bulk water in the whole frequency region. It has been considered that vibrational relaxation of HB OH of bulk water ($H_2O$) proceeds by passing its energy to the HOH bend overtone that is in the Fermi resonance through the anharmonic intramolecular coupling[8,53,57]. In fact, IR pump-probe experiments on bulk water showed that the $T_1$ time of the OH stretch markedly slows down ($T_1$ ~ 700 fs on average) and exhibits

frequency dependence upon isotopic dilution[58], which is explained by the lack of efficient anharmonic coupling between the OH stretch and the HOD bend overtone in a HOD molecule. The high similarity of the $T_1$ time of HB OH stretch between the air/water (H$_2$O) interface and bulk water (H$_2$O) strongly suggests that the vibrational relaxation of the HB OH stretch at the air/water interface also predominantly proceeds with intramolecular anharmonic coupling between the HB OH stretch and HOH bending overtone. Actually, this vibrational relaxation mechanism can rationalize the high similarity of the $T_1$ time between the air/water interface and bulk water, even though the interfacial water is considered to experience a significantly different environment at the molecular level. It is because if the anharmonic coupling between the HB OH stretch and HOH bend overtone governs vibrational relaxation of the HB OH stretch, it is basically an intramolecular property of a water molecule and is expected not to be very sensitive to the difference in the environment. Experimental[5–7] and theoretical studies[55,59] indicated that the HB OH stretch is delocalized over ~10 water molecules in bulk water, whereas at the air/water interface, several studies suggested a much smaller delocalization of the HB OH stretch[60,61]. The similar $T_1$ time between the air/water interface and bulk water implies that the delocalization of the HB OH stretch does not significantly affect the vibrational relaxation process, which is consistent with our conclusion. In a previous study that claimed slow HB OH vibrational relaxation at the air/water interface, it was mentioned that a lower water density at the air/water interface can give rise to the slow vibrational relaxation if the intermolecular Förster resonance energy transfer governs the vibrational relaxation at the interface[42]. The presence of the slower vibrational relaxation at the air/water interface is denied by the present measurements, implying that the lower water density at the interface does not noticeably affect the $T_1$ time of the HB OH. This also supports our argument that the relaxation mechanism of HB OH at the interface has an intramolecular origin.

The frequency of HOH bend overtone is considered to be ~3300 cm$^{-1}$. Thus, the HB OH stretch at around this frequency is expected to strongly be coupled with the HOH bend overtone. However, the ultrafast fluctuation of the hydrogen bonding of water induces very rapid spectral diffusion that shuffles the frequency of the HB OH[5–7,51,62–64]. Actually, it has been established that spectral diffusion of the HB OH stretch proceeds on a time scale of ~50–200 fs in bulk water[5–7,51]. Also at the air/water interface, previous TR HD-VSFG and 2D HD-VSFG studies indicated that the spectral diffusion proceeds on a similar time scale, although the time-resolution of the interfacial measurements was limited (~200 fs)[34,43]. Therefore, the ultrafast spectral diffusion can wash out the memory of the initial frequency of the HB OH stretch, making its $T_1$ time insensitive to the excitation frequency. The possible slight increase of the $T_1$ time on the high-frequency side (Fig. 5) might reflect the remained memory of the initial excitation frequency that is distant from the frequency of the HOH bend overtone, i.e., the funnel of the vibrational relaxation of the excited HB OH. To confirm ultrafast spectral diffusion at the air/water interface, we performed a 2D HD-VSFG measurement for the ground-state bleach region of the HB OH stretch. The 2D HD-VSFG spectrum obtained at 400 fs shows that the ground-state bleach lobe is almost completely elongated along the pump frequency axis, showing that the time scale of the spectral diffusion at the air/water interface is shorter than or, at least, comparable to the $T_1$ time determined in the present study. (See Supplementary Note 6 in Supplementary Information for the details.) The 2D HD-VSFG data support our argument that the $T_1$ time of HB OH stretch is insensitive to the pump frequency because of the ultrafast spectral diffusion.

In our previous study, we showed that the difference between the thermalized signals observed with excitations at 3300 and 3500 cm$^{-1}$ becomes larger as the IR pump pulse energy increases[43]. This indicates that the local quasi-equilibrated temperature after vibrational

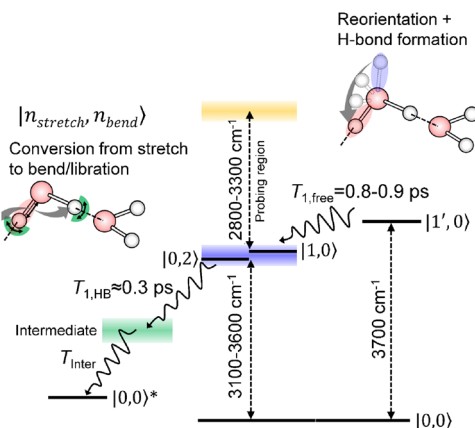

**Fig. 6 | Schematic of the vibrational relaxation process of OH stretch at the air/water (H$_2$O) interface.** States $|1',0\rangle$, $|1,0\rangle$, and $|0,2\rangle$ indicate the first excited state of the free OH stretch, the first excited state of HB OH stretch, and the second excited state (overtone) of HOH bending, respectively. The state $|0,0\rangle^*$ corresponds to a hot ground state of water molecules in which low-frequency vibtations are excited. Blue and red shadings on the OH bond of the schematic water molecule in the upper right indicates the excited free- and excited HB OH stretch vibrations, respectively.

relaxation at the air/water interface can be noticeably different depending on the pump frequency, in the case of the excitation with high pump pulse energy. In fact, this gave rise to the long-lasting (>1 ps) feature in 2D HD-VSFG spectra measured with ~100 μJ IR pump pulses, which was misinterpreted as exceptionally slow spectral diffusion in a previous study[41]. The moderate pulse energy (~15 μJ) of the IR pump adopted in the present and our previous TR-HD-VSFG experiment[43] minimizes the difference in the local temperature for different excitation frequencies and enables us to observe genuine vibrational relaxation and ultrafast spectral diffusion.

The present TR-HD-VSFG study on the excited-state HB OH band region enabled us to determine reliable vibrational relaxation $T_1$ time of the OH stretch at the air/water interface in the whole frequency region, including that of the free OH stretch. Combining the results obtained in this study with those of our previous studies on the vibrational relaxation of the free OH/OD[44,46], we can provide a unified picture of the vibrational relaxation of the OH stretch at the air/water interface, as depicted in Fig. 6.

What makes the vibrational relaxation of OH stretch at the air/water interface distinct is the free OH at ~3700 cm$^{-1}$, which exists only at the interface. As our previous TR-HD-VSFG measurement showed[44], the excitation of the free OH generates the GSB of the free OH at ~3700 cm$^{-1}$ with the free OH excited-state band at around 3550 cm$^{-1}$. The GSB and excited-state band of the free OH simultaneously disappear with a time constant of ~0.9 ps at the air/water interface. The same $T_1$ time of the excited-state free OH was observed with 50% and 20% isotopic dilution within the error[44]. On the other hand, the $T_1$ time of the excited-state free OD stretch is twice larger than the $T_1$ time of the excited-state free OH with no isotopic dilution effect[46]. The no isotopic dilution effect and large isotopic substitute effect on the free OH $T_1$ time reveals that the vibrational relaxation of the excited free OH proceeds with reorientational motion that converts free OH to HB OH at the interface, because a significant isotopic dilution effect is anticipated for the vibrational relaxation through energy transfer. The loss of the excited free OH is concomitantly compensated with the appearance of an unexcited free OH that is generated from either an unexcited HB OH or an unexcited free OH lying almost horizontally by rotation, as the total number of the free OH should be unchanged under the equilibrium condition. Therefore, the repopulation of the unexcited

free OH and the disappearance of the excited free OH are observed with the same time constant.

In the present HD-VSFG study, it was observed that the excited-state HB OH appears with excitation of the free OH with a time constant of $0.84 \pm 0.08$ ps. This time constant is essentially the same as the $T_1$ time of the excited free OH ($0.87 \pm 0.06$ ps)[44] that was determined from the recovery of GSB and the decay of the excited-state band of the free OH. Therefore, the observed excited-state HB OH is attributed to that converted from the excited free OH by the reorientation. In other words, the relaxation of the excited free OH proceeds with the reorientation motion of one OH moiety of a topmost water molecule while keeping its vibrational excitation. The reorientation time of the free OH is shorter[44,46] than the "jump" motion time of the OH moieties in bulk water[65], which is determined from the potential barrier between two stable hydrogen bond configurations[66]. A molecular dynamic simulation indicated that the potential energy of free OH at the air/water interface smoothly increases as the polar angle of orientation changes from 0° (H-up) to 180° (H-down)[67]. Different from the bulk water case, this barrierless potential profile at the interface makes the orientational motion of the free OH proceed in a "diffusive" manner, and the reorientation of the free OH does not require any breaking-formation process of hydrogen bonds[56,67–69]. The rotated excited OH forms a hydrogen bond with the nearby water molecule to generate the excited HB OH as observed in the present study. This hydrogen-bonding formation process is considered to occur almost instantaneously because the time constant of the appearance of the excited-state HB OH is the same as the $T_1$ time of the free OH that is determined by the disappearance of the excited-state free OH[44]. Previous 2D IR experiments of bulk water with theoretical calculations found that the water molecules in nonhydrogen-bonded configuration turn into the hydrogen-bonded configuration in very short time (~100 fs)[66,69]. Therefore, if the surrounding environment of the excited free OH after the reorientational motion is more or less similar to the environment of bulk water, its conversion to the excited HB OH is expected to occur faster than the time resolution of the present measurements (~200 fs). For the rotated excited free OH to form a stable hydrogen-bonded configuration immediately, a partner water molecule acting as a hydrogen bond acceptor needs to approach the rotated free OH and realize a geometrical condition required. Therefore, the reorientation of the free OH is highly likely accompanied by a concurrent motion of the adjacent partner water molecule. The present study also shows that the $T_1$ time of the converted excited-state HB OH is $0.34 \pm 0.06$ ps, which is essentially the same as the $T_1$ time of the excited-state HB OH generated with direct excitations of the HB OH. It implies that the vibrational relaxation of the excited HB OH converted from the excited free OH is indistinguishable from that of excited HB OH generated directly by photoexcitation.

The excited HB OH, whether it is directly excited or converted from the excited free OH, lies in a broad energy continuum (3100–3600 cm$^{-1}$) reflecting the inhomogeneity of hydrogen bonding strength of water. The vibrational relaxation of the excited HB OH at the air/water interface is very similar to that in bulk water, which indicates that it is predominantly governed by the intramolecular anharmonic coupling with HOH bending overtone. This relaxation mechanism is basically of intramolecular origin, so the relaxation dynamics of the excited HB OH does not exhibit a noticeable difference from bulk although the water molecule experiences a substantially different environment at the interface. Although the relevant coupling strength depends on the frequency of the HB OH stretch because of the frequency difference from the HOH bending overtone, the vibrational relaxation $T_1$ time of excited HB OH is not sensitive to its frequency. This insensitive frequency dependence can be rationalized by the ultrafast spectral diffusion induced by the ultrafast fluctuation of the hydrogen bonding network. As indicated by previous TR-HD-VSFG[43] and 2D HD-VSFG studies[34] as well as the 2D HD-VSFG spectrum

obtained in this study (Supplementary Fig. 8 in Supplementary Information), the spectral diffusion process proceeds on a time scale of a few hundred femtoseconds and largely shuffles the frequency of the HB OH stretch. This time scale of the OH stretch frequency redistribution is shorter than or is similar to the time scale of vibrational relaxation of excited HB OH, making its $T_1$ time rather insensitive to the initial frequency of the HB OH stretch. The excited state of HB OH is relaxed through the intermediate state, including the HOH bend and libration, and finally dissipates its excess energy to the surroundings to generate the hot ground state that gives rise to the thermalized signal in the time-resolved spectra at the late delay times. The ~0.9 ps rise time was obtained for the thermalized signal by the SVD analysis (See Supplementary Note 5 in Supplementary Information). The much larger rise time of the thermalized signal than $T_1$ of HB OH stretch determined in the present TR-HD-VSFG study provides experimental evidence of the involvement of the intermediate state in the vibrational relaxation process of the OH stretch at the air/water interface[44].

In summary, the vibrational relaxation $T_1$ time of the OH stretch vibration of water at the air/water (H$_2$O) interface has been determined in the whole OH stretch region, by monitoring the excited-state HB OH band using TR-HD-VSFG spectroscopy. The obtained $T_1$ time of the HB OH stretch ranges between 240 and 340 fs, which is very similar to that of bulk water. This result suggests that the vibrational relaxation of excited-state HB OH at the air/water interface proceeds with an intramolecular relaxation mechanism, i.e., anharmonic coupling with HOH bend overtone, as in the case of bulk water. It was also found that the $T_1$ time of the excited HB OH is not very sensitive to its frequency. This observation is rationalized with the ultrafast spectral diffusion due to ultrafast fluctuation of the hydrogen bonding at the interface. Thanks to the absence of the disturbance from the thermalized signal in the probed region, the present TR-HD-VSFG measurements were able to also detect the appearance of the excited HB OH that is converted from the excited free OH with a time constant of ~0.8–0.9 ps, providing direct experimental evidence for the conversion from an excited free OH to an excited HB OH. These experimental observations give a unified picture of the vibrational relaxation process of OH stretch vibration at the air/water interface, providing a solid basis for understanding the energy dissipation processes at various aqueous interfaces with a truncated hydrogen bonding network.

## Methods

### Time-resolved heterodyne-detected vibrational sum frequency generation (TR-HD-VSFG) spectroscopy

The TR-HD-VSFG setup used in this study has been described elsewhere[40]. Briefly, the light source is a femtosecond Ti:Sapphire regenerative amplifier (Spitfire Ace®, Spectra Physics). The pump IR beam having a bandwidth of ~150 cm$^{-1}$ ($\omega_{pump}$) was focused on the air/water interface. For probing, two probe beams, one centered at 795 nm with a narrow bandwidth ($\omega_{vis}$; ~24 cm$^{-1}$) and the other at 3.3 μm with broad bandwidth ($\omega_{IR}$; ~300 cm$^{-1}$), were first focused on a 10 μm thickness y-cut quartz crystal to generate a local oscillator (LO) pulse, and then, the probe $\omega_{vis}$ and $\omega_{IR}$ beams and the LO were refocused on the air/water interface and spatially overlapped with the pump beam. The time delay (T) of the probe $\omega_{vis}$ and $\omega_{IR}$ pulses with respect to the $\omega_{pump}$ was scanned by a motorized mechanical stage. The generated sum-frequency (SF) signal and LO passed a 750-nm short pass filter and then were introduced to a polychromator (Princeton Instruments, SP-2300i) and detected by a nitrogen-cooled CCD (Princeton Instruments, 10:2KB-dll). Polarization of the SF, probe $\omega_{vis}$, probe $\omega_{IR}$, and pump $\omega_{pump}$ were S-, S-, P-, and P-polarization, respectively. The pulse energies of the probe $\omega_{vis}$ and $\omega_{IR}$ beams were ~7 μJ and ~20 μJ, respectively. The pulse energy of the pump $\omega_{pump}$ beam was 12–15 μJ for the HB OH stretch excitation (3200–3600 cm$^{-1}$) and ~20 μJ for the free OH stretch excitation (3700 cm$^{-1}$). The instrumental response was evaluated from the cross-correlation between

probe $\omega_{IR}$ and pump $\omega_{pump}$ pulses, and it was in the range of $157 \pm 1$ - $211 \pm 9$ fs (FWHM of the fitted Gaussian curves), depending on the pump $\omega_{pump}$ frequency (See Supplementary Fig. 1 of the Supplementary Information).

Ultrapure water from Millipore® dispenser filled in a glass petri dish (3.2 cm diameter) was used for the sample. To prevent the meniscus change due to evaporation, additional water was injected by a syringe pump when the volume height of the water sample was lowered by 200 µm compared to the value at the beginning of the experiment. The surface height was continuously monitored by an optical sensor (Keyence®), and it was maintained within ±1 µm accuracy by a motorized stage during the TR-HD-VSFG experiments. All the TR-HD-VSFG experiments were performed at room temperature (~298 K).

## Reporting summary

Further information on research design is available in the Nature Portfolio Reporting Summary linked to this article.

## Data availability

The data supporting the findings of this study are available within the article or Supplementary Information. Source data are provided with this paper. Additional raw data are available from the corresponding author upon request. Source data are provided with this paper.

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

## Acknowledgements

All the authors (W.S., K.I., S.N., and T.T.) in the present study were supported by JSPS KAKENHI Grant number 18H05265 and 23H00292

## Author contributions

W.S., S.N. and T.T. designed the research. W.S. performed the experiments and analyzed the data. K. I. contributed to the initial part of the experiment. All authors discussed the results and wrote the manuscript.

## Competing interests

The authors declare no competing interests.
