## [Peer Review File · Nature Communications]

Unified Picture of Vibrational Relaxation of OH Stretch at the Air/Water InterfaceREVIEWER COMMENTS

Reviewer #1 (Remarks to the Author):

This manuscript describes TR-HD-SFG measurements of the OH stretch at the water-air interface in order to obtain the T1 relaxation time.

This is in general an interesting topic that is fraught with controversies in the literature.

The new aspect of the present study is that the induced absorption from the 1 to 2 transition is investigated as opposed to previous efforts that have focused on the bleach of the 0 to 1 transition. This gives rise to a cleaner analysis than previous studies.

The experiments are carefully performed and the analysis solid.

However, I don't think that the advancement of our understanding of the relaxation dynamics of the air-water interface achieved in present study is not enough to warrant publication in Nature Communications, or any of the other Nature journals for that matter.

Reviewer #2 (Remarks to the Author):

In this manuscript Sung et al. used heterodyne detected sum-frequency generation spectroscopy (HD-SFG) to measure the rates of OH stretch vibrational relaxation in at water/air interface. This paper makes an important claim that the OH stretch vibrational lifetimes at the interface are comparable to those of bulk water which is in stark contrast to the previously published results. The authors claim that the improvement in experimental technique (HD-SFG vs homodyne SFG) and the "removal" of thermalized signal helped them measure the OH stretch relaxation better. While the authors has made a great effort in convincing the reader that they results are trustworthy, I am hesitant to recommend this paper for publication for the reasons described below.

- As authors claim the mechanism for OH stretch relaxation at the interface is through Fermi resonance which is indeed an accepted view in the water community. The problem is that for Fermi resonance to be efficient the OH stretch frequency needs to be close to ($2 \times \text{HOH bend frequency} - \text{anharmonicity}$) which

is approximately 3250 cm⁻¹. The authors are well-aware of this. This frequency is far away from the free OH stretch frequency of 3700 cm⁻¹. Even if as the authors suggest the reorientation of water occurs such that the free OH get submerged into the bulk water, makes a hydrogen bond, lowering its frequency to enable for the efficient Fermi resonance - all these processes occur on long time scales, definitely longer than 0.5 ps or so. The authors did not provide their estimates for these time scales. Therefore I cannot see how the free OH stretch relaxation can be comparable or even faster. The authors should reconsider their kinetic models for potentially missing contributions.

- The authors rationalize some of their result by referring to spectra diffusion, but no experimental results illustrating OH stretch spectral diffusion at interface are given. I am wondering if two-dimensional SFG measurements can be done. If yes, and their results support the funding of this study than I would be more than happy to recommend this paper for publication in Nature Comm. In its present form the manuscript might be suitable for a more specialized journal.

Minor comments:

- Have the spectra in Fig. 1 been corrected for Fresnel factors?

- Page 16: "In reality..." does this refer to experiments or modeling reality?

Reviewer #3 (Remarks to the Author):

The authors have sought to determine the vibrational relaxation of the hydrogen bonded (HB) OH vibrations of interfacial water, absent any contributions from thermalization or ground state bleach. To do this, they measured the T1 time for the 1- \rightarrow 2 excited state of the HB-OH stretch while pumping on the 0- \rightarrow 1 transition of the HB-OH and free-OH stretch. Their work has shown the T1 time is unaffected by the interfacial bonding environment and, thus, is the vibrational relaxation is the result of an intramolecular relaxation mechanism.

Overall I found this work to be well executed, nicely communicated and of very high quality. It is suitable for publication in Nature Communications. There is, however, one point that it would be nice to hear the authors address.

1. The authors highlight, several times, that the discrepancy between their work and previous work by the Bonn group (which measured the 0- \rightarrow 1 HB-OH transition) is the result of a thermalization of the system. Is it possible for the authors to experimentally measure the dynamics of the thermalization process, to experimentally confirm its contribution? The authors determine a thermalization decay via

the SVD, however it is noted that the error in the SVD determined values for the T_{th} is quite high (100% in some cases). Additionally, with respect to the SVD analysis, the error in some of the T_1 times are extraordinarily high (e.x. $T_{1,free}$ for the 3700 cm^{-1} pump). Should the T_{th} times, in general, be considered with confidence?

Minor Comment:

1. In line 9 of the abstract "HB-OB" should be "HB-OH".

Reply to Reviewer #1:

This manuscript describes TR-HD-SFG measurements of the OH stretch at the water-air interface in order to obtain the T₁ relaxation time.

This is in general an interesting topic that is fraught with controversies in the literature.

The new aspect of the present study is that the induced absorption from the 1 to 2 transition is investigated as opposed to previous efforts that have focused on the bleach of the 0 to 1 transition. This gives rise to a cleaner analysis than previous studies.

The experiments are carefully performed and the analysis solid.

We thank reviewer #1 for highly evaluating the experiments and analysis that we report in this paper. As the reviewer commented, we are very confident that the present study provided the most reliable T₁ values of the OH stretch at the air/water interface.

However, I don't think that the advancement of our understanding of the relaxation dynamics of the air-water interface achieved in present study is not enough to warrant publication in Nature Communications, or any of the other Nature journals for that matter.

Reply:

Although the reviewer wrote that the advancement of our understanding provided by this paper is insufficient to warrant publication in Nature Communications, there are strong reasons that allow us to object to his/her comments. As we write in the introduction, molecular-level properties and dynamics of interfacial water are crucial for understanding many important chemical/physical processes in nature and industry, such as the formation of marine aerosol, on-water catalysis, protein folding, ice formation, functions of supramolecules in the aqueous phase, so on. The energy dissipation process is one of the most fundamental dynamics of interfacial water, and the T₁ time of water OH stretch vibration is a prototypical measure of excess energy dissipation at interfaces. We stress

that the T_1 time of the OH stretch vibration even at the air/water interface is still very controversial although it is the simplest energy dissipation at the most fundamental water interface. Therefore, the unambiguous T_1 time and the unified picture of the vibrational relaxation pathways provided by the present TR-HD-VSFG study provide a foundation for our solid understanding of the energy dissipation process at the liquid interfaces and the starting point for elucidating and controlling various interfacial phenomena. In fact, two other reviewers appreciate the significance of this work.

Furthermore, responding to the comments of other reviewers, we have revised the manuscript by adding new experimental data to Supplementary Information to show 1) ultrafast spectral diffusion of the hydrogen-bonded (HB) OH stretch is almost completed on the time scale of its T_1 at the air/water interface, and 2) experimental evidence confirming that only excited-state HB OH signal appears in the low-frequency region investigated in the present study. With these additional experimental data, the quality and impact of the present paper became even greater than the original manuscript.

Therefore, we strongly believe that this paper meets the requirements for publication in Nature Communications because it provides decisive new knowledge about one of the most fundamental dynamics of interfacial water.

Reply to Reviewer #2:

In this manuscript Sung et al. used heterodyne detected sum-frequency generation spectroscopy (HD-SFG) to measure the rates of OH stretch vibrational relaxation in at water/air interface. This paper makes an important claim that the OH stretch vibrational lifetimes at the interface are comparable to those of bulk water which is in stark contrast to the previously published results. The authors claim that the improvement in experimental technique (HD-SFG vs homodyne SFG) and the "removal" of thermalized signal helped them measure the OH stretch relaxation better. While the authors has made a great effort in convincing the reader that they results are trustworthy, I am hesitant to recommend this paper for publication for the reasons described below.

We thank reviewer #2 for the careful reading and constructive comments. For the original manuscript, the reviewer was hesitant to recommend the publication for two major reasons: one related to the T_1 time of the free OH stretch and the other related to the lack of experimental evidence of the ultrafast spectral diffusion which suppresses the frequency dependence of the T_1 time. Carefully reading the reviewer's comments, we found that the first concern arose from a misunderstanding. To avoid such confusion, we made relevant changes in the revised manuscript. As for the second concern, we performed additional 2D experiments as suggested by the reviewer. The new data clearly indicated spectral diffusion is almost completed on the time scale of the T_1 time. We believe these revisions with the additional data solve the reviewer's concerns.

We describe the details of our responses below.

Comment #1:

- As authors claim the mechanism for OH stretch relaxation at the interface is through Fermi resonance which is indeed an accepted view in the water community. The problem is that for Fermi resonance to be efficient the OH stretch frequency needs to be close to ($2 \times \text{HOH bend frequency} - \text{anharmonicity}$) which is approximately 3250 cm^{-1} . The

authors are well-aware of this. This frequency is far away from the free OH stretch frequency of 3700 cm⁻¹. Even if as the authors suggest the reorientation of water occurs such that the free OH get submerged into the bulk water, makes a hydrogen bond, lowering its frequency to enable for the efficient Fermi resonance - all these processes occur on long time scales, definitely longer than 0.5 ps or so. The authors did not provide their estimates for these time scales. Therefore I cannot see how the free OH stretch relaxation can be comparable or even faster. The authors should reconsider their kinetic models for potentially missing contributions.

Reply:

Reading this comment of the reviewer, we realized that Figure 5 might have made him/her misunderstand the meaning of two T₁ times (T_{1,HB} and T_{1,free}) obtained from the measurement with free OH excitation (3700 cm⁻¹).

First, we would like to make it clear that we analyzed the relaxation of the excited free OH with the following relaxation scheme (i.e. formula (3) in the main text):

Here, the signal observed in this study is that of excited HB OH (HB OH*). Based on this scheme, we described the relaxation process of the excited free OH stretch on page 23, line 1 as follows,

“In the present HD-VSFG study, it was observed that the excited-state HB OH appears with excitation of the free OH with a time constant of 0.84 ± 0.08 ps. This time constant is essentially the same as the T₁ time of the excited free OH (0.87 ± 0.06 ps)⁴⁴ that was determined from the recovery of GSB and the decay of the excited-state band of the free OH. Therefore, the observed excited-state HB OH is attributed to that converted from the excited free OH by the reorientation. In other words, the relaxation of the excited free OH proceeds with the reorientation motion of one OH moiety of a topmost water molecule while keeping its vibrational excitation.”

As clearly written above, the conversion time of an excited free OH stretch to an excited HB OH stretch is $1/k_{1, \text{free}} (T_{1, \text{free}}) = \sim 0.8\text{-}0.9$ ps, and then the generated excited HB OH stretch, which has a much lower frequency, relaxes as an excited HB OH with $1/k_{1, \text{HB}} (T_{1, \text{HB}}) = \sim 0.3$ ps. Thus, the total time for the excited free OH stretch to dissipate its energy is $\sim 1.1\text{-}1.2$ ps.

To avoid any confusion, we note a peculiar feature of this relaxation scheme, that is, the generation time is the excited HB OH from the excited free OH ($T_{1, \text{free}} = \sim 0.8\text{-}0.9$ ps) *is longer than its decay time* ($T_{1, \text{HB}} = \sim 0.3$ ps). In other words, once excited HB OH is formed from the excited free OH “slowly” by a reorientation, the generated excited HB OH relaxes very rapidly with a time constant of 0.3 ps. We stress that, nevertheless, the excited-state HB OH signal grows with the decay time of excited-state HB OH ($T_{1, \text{HB}}$), while the excited-state HB OH signal decays with the formation time of excited-state HB OH ($T_{1, \text{free}}$), as shown in Figure 4(b). This very counter-intuitive temporal behavior of the signal (and the population) simply arises from the relaxation scheme with $T_{1, \text{free}} > T_{1, \text{HB}}$, as we wrote in the 2nd paragraph on page 17.

We are afraid that Figure 5 might have made the reviewer misunderstand this point and think that some portion of the excited free OH relaxes with $T_{1, \text{HB}} = \sim 0.3$ ps. To avoid such confusion, we revised Figures 5 and 6 in the revised main text to emphasize that the total relaxation time of the free OH stretch corresponds to the sum of the excited free OH \rightarrow excited HB OH conversion time ($T_{1, \text{free}} = \sim 0.8\text{-}0.9$ ps) and relaxation time of the generated excited HB OH ($T_{1, \text{HB}} = \sim 0.3$ ps).

Revision:

Figure 5. T₁ time of the HB OH (red circle) and the free OH stretch (blue square) at the air/H₂O interface determined by the decay of the excited-state HB OH stretch signal. The red solid line is an eye guide. T₁ times of bulk H₂O from ref. 42 (green circle) and ref. 6 (green bars) are also plotted for comparison. Black curved arrows connecting the blue square and red circle represent the conversion from the excited free OH to the excited HB OH stretch, which is observed with excitation at 3600 and 3700 cm⁻¹.

Figure 6. Schematic of the vibrational relaxation process of OH stretch at the air/H₂O interface. State $|1',0\rangle$, $|1,0\rangle$, and $|0,2\rangle$ indicate the first excited state of the free OH stretch, the first excited state of HB OH stretch, and the second excited state (overtone) of HOH bending, respectively. The state $|0,0\rangle^*$ corresponds to a hot ground state of H₂O molecules that have weaker hydrogen bonding compared to the steady state. See the main text for the detail.

Comment #2:

- The authors rationalize some of their result by referring to spectra diffusion, but no experimental results illustrating OH stretch spectral diffusion at interface are given. I am wondering if two-dimensional SFG measurements can be done. If yes, and their results support the funding of this study than I would be more than happy to recommend this paper for publication in Nature Comm. In its present form the manuscript might be suitable for a more specialized journal.

Reply:

We thank reviewer #2 for the suggestion of 2D HD-VSFG measurement. According to the reviewer's suggestion, we have measured the 2D HD-VSFG spectrum ($2D \Delta \text{Im}\chi^{(2)}$)

Figure R1. (a) 2D HD-VSFG spectrum at the air/water interface measured at 400 fs. (b) 1D slices of the 2D HD-VSFG spectrum at $\omega_p = 3500 \text{ cm}^{-1}$ (top), 3400 cm^{-1} (middle), and 3300 cm^{-1} (bottom).

spectrum) of the air/water interface at 400 fs (Figure R1a). The observed bleach lobe of the HB OH stretch is vertically elongated along the pump frequency axis. In addition, the bandwidths of the bleach upon excitations at 3300, 3400, and 3500 cm^{-1} (Figure R1b) of the 1D slices are broad ($\sim 200 \text{ cm}^{-1}$ FWHM) and look similar to each other although we can notice a slight difference in the spectral profile of the slice at highest pump frequency $\omega_{\text{pump}} = 3500 \text{ cm}^{-1}$. This newly measured 2D HD-VSFG spectrum indicates that the spectral diffusion of the HB OH stretch at the air/water interface is almost completed on the time scale of the vibrational relaxation of the HB OH stretch which was determined in the present study. Therefore, ultrafast spectral diffusion is the most reasonable rationalization for the insensitivity of frequency dependence of the efficient vibrational relaxation of the HB OH stretch at the air/water interface.

To describe the results of this new 2D HD-VSFG experiment, we revised the main text and added a new section in Supplementary Information.

Revision:

(Main text, Page 20, line 25)

“To confirm ultrafast spectral diffusion at the air/water interface, we performed a 2D HD-VSFG measurement for the ground-state bleach region of the HB OH stretch. The 2D HD-VSFG spectrum obtained at 400 fs shows that the ground-state bleach lobe is almost completely elongated along the pump frequency axis, showing that the time scale of the spectral diffusion at the air/water interface is shorter than or, at least, comparable to the T1 time determined in the present study. (See Section S6 in Supplementary Information for the details.) The 2D HD-VSFG data support our argument that the T1 time of HB OH stretch is insensitive to the pump frequency because of the ultrafast spectral diffusion.”

(Supplementary Information, Page S19, line 1)

S6. 2D HD-VSFG spectrum in the ground-state bleach region at 400 fs

“To confirm the ultrafast spectral diffusion of the HB OH stretch at the air/water interface, we performed a 2D HD-VSFG measurement for the ground-state bleach region of the HB OH stretch (3200 - 3550 cm⁻¹) at 400 fs. The details of our 2D HD-VSFG setup and measurements have been reported elsewhere (Ref. 40 in the main text). Briefly, we measured $\Delta\text{Im}\chi^{(2)}$ spectra using five IR pump frequencies (ω_{pump}) at 3200 cm⁻¹, 3300 cm⁻¹, 3400 cm⁻¹, 3500 cm⁻¹, and 3600 cm⁻¹, and a 2D HD-VSFG spectrum (2D $\Delta\text{Im}\chi^{(2)}$ spectrum) was constructed by combining five $\Delta\text{Im}\chi^{(2)}$ spectra with interpolation along the ω_{pump} axis (y-axis). It is noted that the bandwidth of the IR pump used in the present measurement is narrower (~120-150 cm⁻¹) than that of the IR pump used in our first 2D HD-VSFG measurements at the air/water interface (~200 cm⁻¹) (Ref. 34 in the main text).

Fig. S8(a) shows the 2D HD-VSFG spectrum obtained. The positive lobe in the 2D spectrum is nearly vertical along the ω_{pump} axis, indicating that spectral diffusion is almost completed at 400 fs. Fig. S8(b) depicts the 1D horizontal slices of the 2D HD VSFG spectrum, which correspond to $\Delta\text{Im}\chi^{(2)}$ spectra observed with the IR pump at 3300 cm^{-1} (bottom), 3400 cm^{-1} (middle), and 3500 cm^{-1} (top). These three spectra show that the bandwidth of the positive bleaching band is similar ($\sim 200 \text{ cm}^{-1}$ FWHM) although a subtle difference is noticed for the spectrum at $\omega_{\text{pump}}=3500 \text{ cm}^{-1}$. Based on this 2D HD-VSFG spectrum at 400 fs, we can safely conclude that the time scale of the spectral diffusion at the air/water interface is shorter than or, at least, comparable to the T_1 time of the HB OH stretch determined in the present study (the main text, Tables S2, and S3). This result strongly supports our argument that ultrafast spectral diffusion largely washes out the pump frequency dependence of the T_1 time of the HB OH stretch at the air/water interface.”

Figure S8. (a) 2D HD-VSFG spectrum at the air/water interface measured at 400 fs. (b) 1D slices of the 2D HD-VSFG spectrum at $\omega_p=3500 \text{ cm}^{-1}$ (top), 3400 cm^{-1} (middle), and 3300 cm^{-1} (bottom).

Comment #3:

Minor comments:

- Have the spectra in Fig. 1 been corrected for Fresnel factors?

Reply:

The steady-state $\text{Im}\chi^{(2)}$ spectrum of the air/water interface shown in the manuscript was not corrected by the Fresnel factor. We calculated the Fresnel factor and confirmed that the Fresnel factor correction does not change the $\Delta\text{Im}\chi^{(2)}$ below 3100 cm^{-1} where the excited-state band of the HB OH stretch appears.

To make this point clear, we revised the main text and added a new section about the Fresnel factor correction in Supplementary Information, as follows.

Revision:

(Main text, Page 11, line 3)

“Note that we did not perform the Fresnel factor correction for this analysis, because it does not affect the $\text{Im}\chi^{(2)}$ spectra below 3100 cm^{-1} (see Section S7 in Supplementary Information).”

(Supplementary Information, Page S21, line 1)

S7. Fresnel factor correction on the $\text{Im}\chi^{(2)}$ and $\Delta\text{Im}\chi^{(2)}$ spectra of the air/water interface

Fig. S9 compares $\text{Im}\chi^{(2)}$ spectra at the air/water interface with and without the correction for the Fresnel factor. For the correction, we adopted the three-layer model (Shen, Y. R., *Fundamentals of Sum-frequency Spectroscopy*, Cambridge University Press (2016)) using two different refractive indices for the interface region: one is the refractive indices of bulk H₂O ($n' = n_{\text{H}_2\text{O}}$) and the other is a value in between the refractive indices of air and bulk H₂O ($n_{\text{air}} < n' < n_{\text{H}_2\text{O}}$) which is estimated by applying the Lorentz model to the interface region (Zhuang et al. *Phys. Rev. B* **59**, 12632 (1999)). Since $n_{\text{H}_2\text{O}}$ in this frequency range significantly changes due to the vibrational resonance of water OH stretch, the modulation of the Fresnel factor of $L_{zz}(\omega_2)$ becomes larger as the used n' value approaches $n_{\text{H}_2\text{O}}$. In fact, the largest effect of the Fresnel factor correction is seen

when n' is set to $n_{\text{H}_2\text{O}}$ (Fig. S9(a)). Nevertheless, the effect of the correction on the $\text{Im}\chi^{(2)}$ spectrum is seen only above 3100 cm^{-1} where the IR absorption due to the OH stretch is significant, and the spectral region between 2900 and 3050 cm^{-1} , where the excited-state signal of the HB OH stretch appears, is not influenced even in this case. For a more realistic refractive index of the interface ($n_{\text{air}} < n' < n_{\text{H}_2\text{O}}$), the $\text{Im}\chi^{(2)}$ spectrum only exhibits a subtle shift of the HB OH stretch band after Fresnel factor correction, and the change below 3100 cm^{-1} is negligible (Fig S9(b)).

Fig. S10 shows time-resolved $\Delta\text{Im}\chi^{(2)}$ spectra measured with 3400-cm^{-1} excitation with and without the Fresnel factor correction. (For the convenience of comparison, we multiplied the magnitude of the $L_{yy}(\omega_{\text{SF}})L_{yy}(\omega_1)L_{zz}(\omega_2)$ making the amplitude of the $\Delta\text{Im}\chi^{(2)}$ preserved.) We made the correction using the three-layer model with two interface refractive indices, as in the case of the steady-state $\text{Im}\chi^{(2)}$ spectrum. Compared to the $\Delta\text{Im}\chi^{(2)}$ spectra without the correction (Fig. S10 (a)), the $\Delta\text{Im}\chi^{(2)}$ spectra after the correction using the two n' values only exhibit subtle changes in the frequency region above 3100 cm^{-1} (Fig.S10 (b) and (c)). Furthermore, the temporal profiles of the $\Delta\text{Im}\chi^{(2)}$ signal integrated from 2900 cm^{-1} to 3050 cm^{-1} show almost no change with the Fresnel factor correction, and the evaluated $T_{1,\text{HB}}$ values are essentially the same (Fig. S11). These results of the analysis show that the effect of the Fresnel factor correction is negligible for the $\Delta\text{Im}\chi^{(2)}$ spectra, temporal traces of the excited-state transition signal of the HB OH stretch, and evaluated T_1 values discussed in the main text.

Figure S9. $\text{Im}\chi^{(2)}$ spectrum at the air/water interface with and without the Fresnel factor correction. (a) $\text{Im}\chi^{(2)}$ spectrum corrected with the interfacial refractive index set to the bulk H₂O value ($n^1=n_{\text{H}_2\text{O}}$). (b) $\text{Im}\chi^{(2)}$ spectrum corrected with the interfacial refractive index estimated by applying the Lorentz model to the interface ($n_{\text{air}} < n^1 < n_{\text{H}_2\text{O}}$). In both panels, $\text{Im}\chi^{(2)}$ spectrum without the correction is also shown for the comparison.

Figure S10. $\Delta\text{Im}\chi^{(2)}$ spectra at the air/water interface with and without the Fresnel factor correction. (a) $\Delta\text{Im}\chi^{(2)}$ spectra without the Fresnel factor correction (same as Fig.2(c) in the main text). (b) $\Delta\text{Im}\chi^{(2)}$ spectra corrected with the interfacial refractive index set to the bulk H_2O value ($n' = n_{\text{H}_2\text{O}}$). (c) $\Delta\text{Im}\chi^{(2)}$ spectra corrected with the interfacial refractive index estimated by applying the Lorentz model to the interface ($n_{\text{air}} < n' < n_{\text{H}_2\text{O}}$).

Figure S11. Temporal traces of the $\Delta \text{Im}\chi^{(2)}$ signal with and with the Fresnel factor correction. The $\Delta \text{Im}\chi^{(2)}$ signal is obtained with the pump frequency at 3400 cm^{-1} . (a) Trace without Fresnel factor correction (same as Fig.3(c) in the main text). (b) Trace after the Fresnel factor correction with the interfacial refractive index set to the bulk H_2O value ($n' = n_{\text{H}_2\text{O}}$). (c) Trace after the Fresnel factor correction corrected with the interfacial refractive index estimated by applying the Lorentz model to the interface ($n_{\text{air}} < n' < n_{\text{H}_2\text{O}}$).

Comment #4:

- Page 16: "In reality..." does this refer to experiments or modeling reality?

Reply:

In the original manuscript, it refers to both the experiment and model. To avoid unnecessary confusion, we remove this phrase on page 16 as follows,

Revision:

(page 16, line 16)

"In the reality, however,..." (previous)

"However,..." (present)

Reply to Reviewer #3:

The authors have sought to determine the vibrational relaxation of the hydrogen bonded (HB) OH vibrations of interfacial water, absent any contributions from thermalization or ground state bleach. To do this, they measured the T1 time for the 1->2 excited state of the HB-OH stretch while pumping on the 0->1 transition of the HB-OH and free-OH stretch. Their work has shown the T1 time is unaffected by the interfacial bonding environment and, thus, is the vibrational relaxation is the result of an intramolecular relaxation mechanism.

Overall I found this work to be well executed, nicely communicated and of very high quality. It is suitable for publication in Nature Communications. There is, however, one point that it would be nice to hear the authors address.

We thank the reviewer for his/her very positive comments. For the question that the reviewer raised, we performed additional experiments to provide a clear answer, as described below.

Comment #1:

1. The authors highlight, several times, that the discrepancy between their work and previous work by the Bonn group (which measured the 0->1 HB-OH transition) is the result of a thermalization of the system. Is it possible for the authors to experimentally measure the dynamics of the thermalization process, to experimentally confirm its contribution? The authors determine a thermalization decay via the SVD, however it is noted that the error in the SVD determined values for the T₁ is quite high (100% in some cases). Additionally, with respect to the SVD analysis, the error in some of the T₁ times are extraordinarily high (e.x. T_{1,free} for the 3700 cm⁻¹ pump). Should the T₁ times, in general, be considered with confidence?

Reply:

We would like to first stress that, for the evaluation of the T_1 time, which is the main objective of the present study, the determination of the accurate T_{th} is not necessary. The SVD analysis described in Supplementary Information was performed to confirm that the basis of the analysis in the present study, that is, the $\Delta Im\chi^{(2)}$ signal in the frequency region lower than $\sim 3100\text{ cm}^{-1}$ arises solely from the excited-state HB OH. For the comparison, we also carried out the SVD analysis for the spectra in a wider region of $2900 - 3250\text{ cm}^{-1}$, which also contains the thermalized signal, and obtained the T_{th} value by the analysis using two spectral components. However, the T_{th} times (as well as some T_1 times) deduced from this SVD analysis are not accurate due to the contribution of the ground-state bleach of the HB OH stretch observed immediately after photoexcitation (particularly for the 3200 cm^{-1} and 3300 cm^{-1} excitation), and because only red edge of the thermalized spectrum is observed in the low-frequency region investigated ($2900 - 3250\text{ cm}^{-1}$). Therefore, it is difficult to determine the accurate T_{th} value from the data obtained in the present study.

Stimulated by this comment of the reviewer, however, we carried out new experiments for the higher frequency region of $2900 - 3550\text{ cm}^{-1}$ where the main spectral feature due to the thermalized signal appears. In this region, the main part of the ground-state bleach of the HB OH stretch which exhibits spectral diffusion also appears. Since the SVD analysis cannot well cope with a gradual spectral change such as spectral diffusion, we cannot apply SVD analysis to the time-resolved $\Delta Im\chi^{(2)}$ spectra in this region when measured with narrow-band IR excitation. Therefore, we employed a broadband IR pump pulse that can excite the entire HB OH stretch band to minimize the spectral diffusion of the bleach band. Furthermore, the present study on the excited-state HB OH signal revealed that the T_1 time of HB OH stretch is rather insensitive to the pump frequency. Thus, the obtained time-resolved $\Delta Im\chi^{(2)}$ spectra were expected to simply be represented by the linear combination of the two spectral components, i.e., (1) ground-state bleach & excited-state HB OH stretch and (2) thermalized signal. Fig. R2 below shows $\Delta Im\chi^{(2)}$ spectra obtained with broadband IR excitation covering the whole HB OH

stretch band at the air/water interface as well as the result of the SVD analysis. As anticipated, the observed time-resolved spectra are well decomposed into the bleach and excited-state band of the HB OH stretch which decays of $T_{1,HB} = 0.34$ ps, and thermalized signal which rises with $T_{th} = 0.85$ ps. Moreover, the spectrum of the thermalized signal obtained with the SVD analysis (Fig R2(c)) does not show any signals below 3100 cm^{-1} , further confirming the assumption of the present study. We note that the obtained T_{th} value (0.85 ps) is substantially larger than the $T_{1,HB}$ (0.3 - 0.4 ps), implying the involvement of the intermediate state of the relaxation process of excited HB OH stretch. This additional experiment provides the most reliable T_{1th} time to date, although we cannot examine excitation frequency dependence of the T_{th} with this broadband IR excitation.

We added a new paragraph in Supplementary Information to describe this new TR-HD-VSFG experiments with broadband IR excitation.

Figure R2. (a) $\Delta \text{Im}\chi^{(2)}$ spectra in the IR probe range of 2900 - 3550 cm^{-1} and the spectral decomposition based on SVD. Experimental data (shaded area), decomposed spectral components due to the excited-state transition + ground-state bleach bands of the HB OH stretch (red line), thermalized signal (blue line), and their sum (black line). The spectral profile of the broadband IR pump is shown on top of the $\Delta \text{Im}\chi^{(2)}$ spectra. (b) Singular values obtained from the SVD analysis. (c) Spectral components due to excited-state and ground-state bleach bands (top) and thermalized signal (bottom). (d) Temporal traces of the two decomposed spectral components.

Revision:

(Supplementary Information, Page S16, line 1)

S5. TR-HD-VSFG experiments using broadband excitation in the ground-state bleach region and singular value decomposition (SVD) analysis

To obtain information about the thermalized signal, we performed TR-HD-VSFG measurements in the ground-state bleach region. In this experiment, we used a broadband IR pump pulse that can excite the whole broad HB OH stretch band to minimize the spectral evolution due to spectral diffusion. The broadband IR pump pulse was generated through self-phase modulation by loosely focusing the narrow-band IR pump pulse ($\omega_{\text{pump}} \sim 3400 \text{ cm}^{-1}$) onto 2 mm YAG (111) and 1 mm silicon plates, and it was spatially filtered by an iris after the collimating lens. Figure S7(a) shows the obtained $\Delta\text{Im}\chi^{(2)}$ spectra, which show both ground-state bleach ($3200\text{-}3550 \text{ cm}^{-1}$) and excited-state HB OH ($2900\text{-}3200 \text{ cm}^{-1}$) bands immediately after IR excitation. The $\Delta\text{Im}\chi^{(2)}$ spectrum exhibits a noticeable spectral change in the sub-picosecond time region. After 1 ps, the $\Delta\text{Im}\chi^{(2)}$ spectra consist mostly of the thermalized signal exhibiting positive and negative spectral features below and above $\sim 3500 \text{ cm}^{-1}$, respectively, which is consistent with our previous TR-HD-VSFG study upon narrowband excitation (Ref. 43 in the main text). As seen, the spectrum of the thermalized signal does not exhibit any signal in the frequency region below 3100 cm^{-1} .

When we perform TR-HD-VSFG measurements in this ground-state bleach region using a narrow-band IR pump pulse, we observe ultrafast spectral diffusion (Ref. 43 in the main text). Thus, we cannot safely carry out SVD analysis to the time-resolved spectra because SVD cannot reasonably treat gradual spectral change such as spectral diffusion. However, the broadband excitation employed in this experiment does not create a spectral hole in the broad HB OH stretch band and hence suppresses spectral diffusion. Furthermore, the present study on the excited-state HB OH signal revealed that the T_1 time of HB OH stretch is rather insensitive to the pump frequency. Thus, the time-resolved $\text{Im}\chi^{(2)}$ spectra obtained with broadband excitation are expected to simply be represented by the linear combinations of the two spectral components, i.e., (1) ground-state bleach

& excited-state HB OH stretch bands and (2) thermalized signal. Therefore, we performed SVD analysis and actually obtained only two major singular values that correspond to these two spectral components (Fig. S7(b)). In the same way as the SVD analysis described in section S4, we applied model fitting using exponential decay and rise to obtain a transformation matrix C , and obtained spectral components and temporal traces of the excited-state population of the HB OH stretch and thermalized signal by using this C (Fig. S7(c) and (d)). The spectral component consisting of the excited-state transition and ground-state bleach of the HB OH stretch decay with a time constant of 0.34 ± 0.05 ps, which accords very well with the T_1 time of the HB OH stretch determined from the temporal traces of the excited-state signal in the low-frequency region in the present study (Fig. 5 of the main text). On the other hand, the rise time of the spectral component corresponding to the thermalized signal is determined to be 0.85 ± 0.28 ps, and this time constant at the air/water interface is very similar to the rise time of the thermalized signal in bulk water (700-800 fs) which was determined by time-resolved IR experiments (Ref. 5 and 6 in the main text).

Figure S7. (a) $\Delta \text{Im}\chi^{(2)}$ spectra in the IR probe range of 2900 - 3550 cm^{-1} and the spectral decomposition based on SVD. Experimental data (shaded area), decomposed spectral components due to the excited-state transition + ground-state bleach bands of the HB OH stretch (red line), thermalized signal (blue line), and their sum (black line). The spectral profile of the broadband IR pump is shown on top of the $\Delta \text{Im}\chi^{(2)}$ spectra. (b) Singular values obtained from the SVD analysis. (c) Spectral components due to excited-state and ground-state bleach bands (top) and thermalized signal (bottom). (d) Temporal traces of the two decomposed spectral components.

Comment #2:

Minor Comment:

1. In line 9 of the abstract "HB-OB" should be "HB-OH".

Reply:

We thank the reviewer for careful reading and for letting us know the typo. We have corrected it.

REVIEWERS' COMMENTS

Reviewer #2 (Remarks to the Author):

I am happy with the revised manuscript. I recommend publication but the following issue needs to be fixed first.

Figure S11. Caption. "signal with and with the Fresnel..." should be "with and without".

Reviewer #3 (Remarks to the Author):

The authors have sufficiently addressed my previous question. The additional experiments carried out to measure the dynamics of the thermalized and the spectral diffusion (in response to another reviewer's question) are welcome additions.

It is my opinion this article is of sufficient quality and significance for publication in Nature Communications, in its current form. The discussion, conclusions, and data sets included in this manuscript are immensely important to resolving inconsistencies in the literature surrounding the dynamics of interfacial water.

I congratulate the authors on a well-executed and thorough study.

Reply to Reviewer #2:

I am happy with the revised manuscript. I recommend publication but the following issue needs to be fixed first.

Figure S11. Caption. "signal with and with the Fresnel..." should be "with and without".

Reply:

According to the reviewer comment, we have corrected the typo in the caption of Figure S11 of Supplementary Information. We thank again reviewer #2 for careful reading.

Reply to Reviewer #3:

The authors have sufficiently addressed my previous question. The additional experiments carried out to measure the dynamics of the thermalized and the spectral diffusion (in response to another reviewer's question) are welcome additions.

It is my opinion this article is of sufficient quality and significance for publication in Nature Communications, in its current form. The discussion, conclusions, and data sets included in this manuscript are immensely important to resolving inconsistencies in the literature surrounding the dynamics of interfacial water.

I congratulate the authors on a well-executed and thorough study.

Reply:

We truly appreciate the reviewer #3's who recommended publication of our paper as it is. Since he did not request any changes, we did not make any revisions on his/her comments.